# Two-photon activated precision molecular photosensitizer targeting mitochondria

Inês F. A. Mariz[1], Sandra N. Pinto [2,3], Ana M. Santiago[1], José M. G. Martinho [1], Javier Recio [4], Juan J. Vaquero[4], Ana M. Cuadro [4✉] & Ermelinda Maçôas [1✉]

Mitochondria metabolism is an emergent target for the development of novel anticancer agents. It is amply recognized that strategies that allow for modulation of mitochondrial function in specific cell populations need to be developed for the therapeutic potential of mitochondria-targeting agents to become a reality in the clinic. In this work, we report dipolar and quadrupolar quinolizinium and benzimidazolium cations that show mitochondria targeting ability and localized light-induced mitochondria damage in live animal cells. Some of the dyes induce a very efficient disruption of mitochondrial potential and subsequent cell death under two-photon excitation in the Near-infrared (NIR) opening up possible applications of azonia/azolium aromatic heterocycles as precision photosensitizers. The dipolar compounds could be excited in the NIR due to a high two-photon brightness while exhibiting emission in the red part of the visible spectra (600–700 nm). Interaction with the mitochondria leads to an unexpected blue-shift of the emission of the far-red emitting compounds, which we assign to emission from the locally excited state. Interaction and possibly aggregation at the mitochondria prevents access to the intramolecular charge transfer state responsible for far-red emission.

[1] Centro de Química Estrutural (CQE) and Institute of Molecular Sciences (IMS), Instituto Superior Técnico, Universidade de Lisboa, 1049-001 Lisboa, Portugal. [2] Institute for Bioengineering and Biosciences (IBB) Instituto Superior Técnico, Universidade de Lisboa, Av. Rovisco Pais, 1049-001 Lisboa, Portugal. [3] Associate Laboratory - Institute for Health and Bioeconomy (i4HB), Instituto Superior Técnico, Universidade de Lisboa, Av. Rovisco Pais, 1049-001 Lisboa, Portugal. [4] Departamento de Química Orgánica y Química Inorgánica, Universidad de Alcalá, (IRYCIS), 28871-Alcalá de Henares, Madrid, Spain. ✉email: ana.cuadro@uah.es; ermelinda.macoas@tecnico.ulisboa.pt

Two-photon excitation is routinely used in multiphoton fluorescence microscopy for three-dimensional (3D) imaging of thick biological samples. Such imaging modality is becoming increasingly relevant given the paradigm shift in preclinical research towards the use of more realistic 3D models that can mimic the complex tissue environment and architecture of real biological systems. The increasing demand for better diagnostic tools, and more efficient targeted therapeutics alongside with the development of innovative nanomaterials with nonlinear optical response have also promoted a renewed interest in exploring two-photon excitation in biomedical applications. Progresses have been reported, not only at the level of fluorescent labels with increased two-photon brightness and improved specificity for subcellular organelles and biologically relevant analytes[1–6], but also at the level of devices for the clinical practice[7,8]. Typically, the two-photon induced process, being it emission, singlet oxygen generation or heat release, is activated by excitation in the NIR-I spectral region (650–950 nm) that corresponds to the transparent optical windows of biological tissue, thus allowing an increased penetration depth of the excitation light[9]. Concomitantly, the quadratic dependence of the two-photon absorption (TPA) probability on the light intensity confines the process to a highly localized focal volume reducing the off-target photodamage.

While nanomaterials such as gold clusters, semiconductor quantum dots and graphene quantum dots have been reported to have very high TPA cross-sections[10–13], the development of nonlinear molecular materials continues to be of high relevance due to the more favourable cellular uptake kinetics and distribution[14]. Subcellular targeting of nanoparticles is challenging due to internalization via endocytosis that often traps them into endosome or lysosome[15]. One of the strategies to design molecules with an efficient two-photon response is based on push-pull architectures with cationic electron acceptor groups. The focus has been on benzothiazolium[16–18] and azinium (pyridinium, quinolinium) cations[19–21]. In our recent work, we have expanded the range of cationic acceptors used in two-photon fluorophores to azonia aromatic heterocycles[22,23]. Azonia salts have been used as probes of biomacromolecules and biologically relevant analytes[24–26]. Their lipophilic cationic nature has a great potential for mitochondria targeting[27].

In the last decade, mitochondria metabolism has emerged as an interesting target for cancer therapy[28,29]. As a result, there was an exponential growth of mitochondria-targeted two-photon

fluorophores[6,30–32]. Nevertheless, the development of mitochondria-targeted two-photon activated molecular photosensitizers (PS) is still lagging behind. Many reports exist of two-photon activated PS lacking the mitochondria targeting ability[33,34], and also on mitochondria-targeted PS activated by linear excitation in the NIR, lacking 3D-precision attainable by nonlinear excitation[35–38]. With few exceptions[39–42], the design of molecular PS combining the two features, mitochondria targeting and two-photon activation, is currently focused on metalated tetrapyrrolic macrocycles and transition metal complexes. The first approach results in molecules with high TPA cross-sections (>1000 GM) and singlet oxygen quantum yields (>0.5), but poor solubility in biological medium. The second approach leads to less hydrophobic compounds at the cost of significantly lower TPA cross-sections (<200 GM)[43].

Here we report an original molecular design for two-photon activated and mitochondria-targeted PS based on dipolar and quadrupolar quinolizinium and benzimidazolium cations. All the studied compounds show a high affinity towards mitochondria. Some of the compounds show an efficient light-induced mitochondria damage upon two-photon excitation in the NIR opening possible applications of azonia/azolium aromatic heterocycles as precision photosensitizers. In addition, interaction with the mitochondria promotes an unexpected blueshift in the emission spectra that is most noticeable in the far-red emitting compounds. Changes in the optical properties upon interaction with subcellular organelles can be explored to provide a higher selectivity in fluorescent labelling. Examples of that are the interaction of dyes with DNA increasing their quantum yields in the nucleus, and aggregation or hindered rotation increasing the quantum yield at the mitochondria[32,44].

## Results and discussion

**Design and synthesis of chromophores Q and B.** Compounds based on quinolizinium (**Q**) and benzimidazolium (**B**) cations have been designed for selective in vivo labelling of subcellular organelles and controlled mitochondria damage. Our goal was to improve the two-photon brightness and selectivity through changes in the molecular structure with emphasis on the nature of the electron acceptor (A) and electron-donating (D) substituents, and the extent of electronic delocalization. We designed simple dipolar molecules (D-π-A⁺) or more complex D-π-A⁺-A⁺-π-D and D-π-A⁺-π-D architectures, as depicted in Fig. 1. The synthetized compounds are referred as **Q** or **B** depending on whether they are quinolizinium or benzimidazolium derivatives with the addition of an "e" for extended quadrupolar architectures (**Qe** and **Be**). The donor groups in the structure are identified by numbers: **2** for [((4-methoxyphenyl) ethenyl) phenyl], **3** for 1H-indole and **4** for 4-dimethylaminophenyl. Using simple/double Knoevenagel condensations, eight compounds were obtained with spectroscopic purity and good yields for which the upconverted emission properties and the performance as labels in live animal cells is here discussed for the first time.

Compounds **B2**, **Be2**, **B4** and **Be4** were obtained via Knoevenagel reaction between the corresponding arylaldehyde derivatives of the donor groups, (4-[(1E)-2-(4-methoxy-phenyl) ethenyl]-benzaldehyde for **B2/Be2** and 4-dimethylamino-benzaldehyde for **B4/Be4**, and either 1,2,3-trimethylbenzimidazolium iodide (**B2** and **B4**) or 1,1′,2,2′, 3,3′-hexamethyl-1H,3′H-5,5′-bisbenzimidazolium iodide (**Be2** and **Be4**). The dipolar quinolizinum derivatives, **Q2** and **Q3**, were synthetized from 2-methylquinolizinium hexafluorophosphate and 4-[(1E)-2-(4-methoxy-phenyl)ethenyl]-benzaldehyde and 1H-indole-3-carboxaldehyde respectively following a similar Knoevenagel procedure to other reported quinolizinium analogous[22,23] as illustrated in

**Fig. 1 Synthesized compounds.** Selected push-pull systems based on either the quinolizinium (Q and Qe) or benzimidazolium (B and Be) cations as electron acceptor cores combined with electron donor groups 2–4.

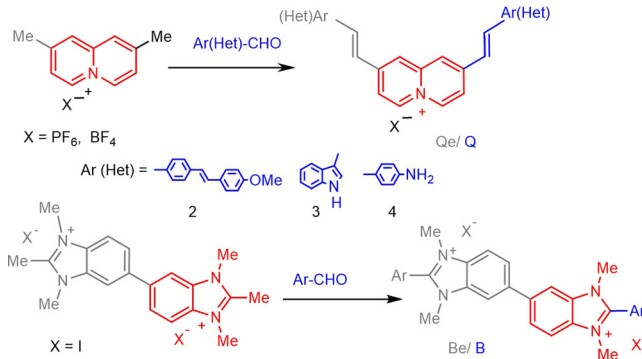

**Fig. 2 General synthetic procedure.** Schematic illustration of the synthesis of quadrupolar and dipolar chromophores Qe/Be and Q/B.

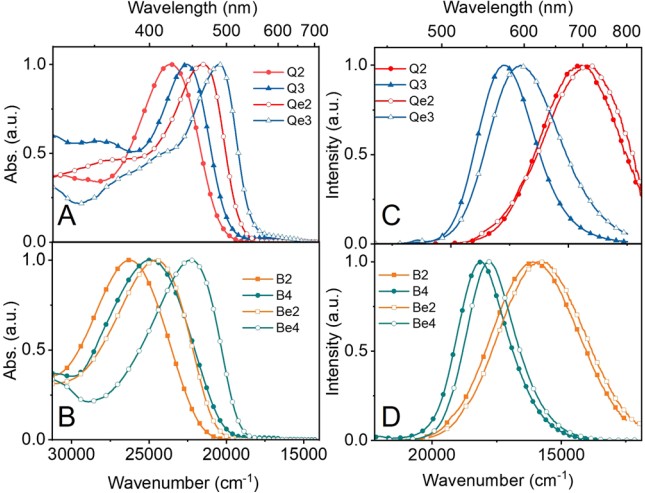

**Fig. 3 Optical properties of the synthesized compounds.** Absorption (A and B) and emission (C and D) spectra of the lipophilic cations in DMSO.

Fig. 2. The synthesis of compounds **Qe2** and **Qe3** was reported earlier[45], but neither the nonlinear optical properties nor the staining of live animal cells was studied.

**Linear optical properties**. The absorption and emission spectra of the synthesized compounds in DMSO are shown in Fig. 3 and the optical properties are summarized in Table S2. Table S2 includes calculated wavelength for the lowest energy transitions and the corresponding percentage contribution of the most important excitation to the configuration interaction (CI) expansion. Calculated data were obtained from Time-Dependent Density Functional Theory (TD-DFT) calculations on the optimized geometry of the cation in DMSO. According to the calculation, the quinolizinum derivatives adopt a planar structure, whereas in the benzimidazolium derivatives the π-bridge is twisted out of the plane of benzimidazolium unit by about 30º, and in the quadrupolar compounds the two benzimidazolium units are twisted with respect to each other by ~40º. Due to the twisted structure of the benzimidazoliums and its effect on the extent of conjugation, this series of compounds absorb at higher energies when compared with analogous quinolizinium compounds. The peak absorption of **Q2** and **Q3** lays at 422 and 443 nm whereas that of **B2** and **B4** are observed at 380 and 400 nm (Table S2), respectively. Likewise, the peak absorption of **Qe2** and **Qe3** at 465 and 488 nm appear red-shifted from those of **Be2** and **Be4** observed at 407 and 448 nm (Table S2), respectively. Nevertheless, in both series of compounds, the UV–Vis

absorption and emission maxima of the quadrupolar compounds are shifted to the red with respect to the analogous dipolar compound due to the increased conjugation length. The trend observed in the absorption spectra are reasonably well predicted by the calculations as shown in the calculated spectra in Fig. S9. The larger Stokes shift and broad emission spectra in the methoxyphenyl substituted compounds (**Q2**, **Qe2**, **B2** and **Be2**) suggest that the excited state involves a significant charge-transfer. The frontier molecular orbitals involved in Frank-Condom transitions as calculated by the TD-DFT methods (Fig. S10) confirm the charge-transfer nature of the lowest energy transition with a significant electronic density shift from the π-conjugated bridge and the electron donor groups to the cationic centre of the molecules. The emission of the synthesized compounds covers the entire visible spectrum up to the NIR (500–800 nm) in solution. The fluorescence quantum yields span two orders of magnitude, from 0.003 in **Qe2** to 0.34 in **B2** with no observed trend with respect to the conjugation length, electron donor strength or multipolar nature of the compounds. The brightest compounds have quantum yields in the range of 20–30% (**Q2**, **B2** and **Be2**).

**Non-linear optical properties**. All the compounds show TPA within the transparent window of biological tissue with TPA maxima between 810 and 890 nm. In Fig. 4 we represent the TPA cross-section ($\sigma_2$) as a bar that extends over the full width at half maxima (FWHM) of the two-photon induced emission band and is centred at the corresponding peak position. The TPA and emission spectra of all the compounds are shown in the supplementary information (Figs. S11–S14). The nonlinear optical properties are summarized in Table S2.

The quadrupolar compounds have larger TPA cross-sections than the dipolar ones. The difference between the two sets of compounds is not additive suggesting the existence of synergistic interbranch electron correlation effects[23,46–48]. For the quinoliziniums, on average the quadrupolar compounds have 5 times larger TPA cross-sections than the dipolar compounds ($\sigma_2$ of 1253/266 GM and 482/92 GM for **Qe2/Q2** and **Qe3/Q3**, respectively). This difference is smaller in the benzimidazoliums where the quadrupolar compounds performed at least three times better than the dipolar analogue ($\sigma_2$ of 189/55 GM and 281/75 GM for **Be2/B2** and **Be4/B4**, respectively). The distortion from planarity in the benzimidazoliums justifies the smaller difference in the TPA cross-section between the dipolar and quadrupolar compounds.

The results obtained for **Q2** and **Qe2** are similar to those reported previously for analogous quinolizinium derivatives albeit their longer conjugation length[23]. The use of different solvents and different TPA cross-section standards can lead to a dispersion of results that precludes direct comparison with the data reported earlier. Solvent polarity has an influence on the TPA cross-section that is difficult to predict. $\sigma_2$ values have been reported to increase with solvent polarity for low polarity solvents and decrease sharply for highly polar solvents[49]. We checked that in the same measurement condition, in DMSO and using Rhodamine 6G as standard, the compounds with the largest conjugation length (**Q2** and **Qe2**) have a TPA cross-section that is 5–10 times larger than analogous quinolizinium derivatives reported earlier.

The largest TPA cross-section is that of compound **Qe2** (1253 GM), which is the quadrupolar quinolizinium with the methoxyphenyl electron donor group and the longer conjugation length. However, due to its low fluorescence quantum yield ($\phi = 0.003$) the TPA brightness ($\sigma_2\phi$) is quite modest (4 GM). The brightest lipophilic cation is its dipolar analogue **Q2** with a TPA cross-section of 266 GM and a quantum yield of 24% giving

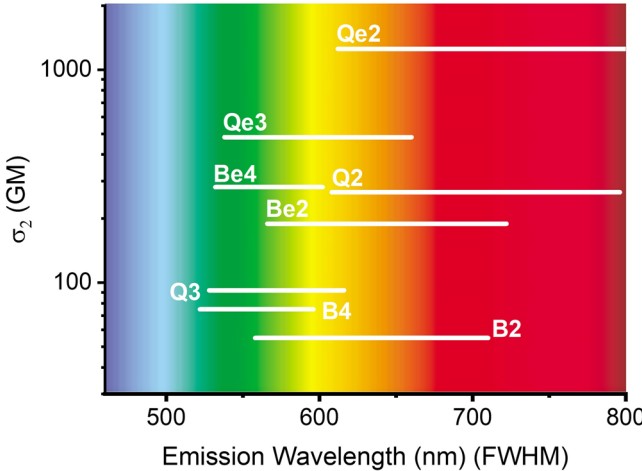

**Fig. 4 TPA cross-section as a function of the emission wavelength.** The width of the bars represents the FWHM.

a nonlinear brightness of 65 GM. Both compounds emit in the red with the emission band centered around 700 nm. With the possibility of NIR excitation and red emission, **Q2** is a good candidate for fluorescent labelling of biological samples. Nevertheless, since interaction with subcellular organelles can affect the performance of the dyes inside the cells, we tested the internalization and labelling of all the compounds in live cultured animal cells.

**Mitochondria targeting**. The performance of the lipophilic cations as fluorescent labels of subcellular organelles in living cells was studied by confocal and multiphoton fluorescence microscopy using HEK 293T cells. No obvious dark toxicity was observed during the experiments for the 2–10 μM concentration range used in this study. HEK 293T cells retained their regular size and shape and the cell nucleus showed, as expected in healthy cells, a smooth spheroid appearance. Compounds **Q2**, **Q3**, **Qe2**, **Qe3**, **B2** and **B4**, showed a sub-cellular distribution typical of mitochondria specific labelling. Mitochondria selectivity was confirmed by a nearly perfect co-localization with the commercial mitochondria selective label MitoTracker Red as shown in Fig. 5.

From left to right we show the emission of the lipophilic cations (green), the mitochondria label MitoTracker Red (red), the nuclear label Hoechst 33342 (blue). The right most panel is the overlay image of the three channels showing in yellow the colocalization of the compounds with the mitochondria label. We checked that there was no crosstalk between the MitoTracker channel (emission in the 600–700 nm range) and the cationic dyes channel (emission in the 500–600 nm range) using control experiments where the cells were incubated only with Mito-Tracker and the nuclear dye (Fig. S15 of Cell culture and staining section of the supplementary information). Fluorescence microscopy images of the cells incubated only with the cationic dyes, using no other fluorescent label, are shown in Fig. S16 upon one-photon excitation. Multiphoton images were also collected for the compounds whose emission was clearly observed inside the cells (Fig. S17). Neither of the quadrupolar benzimidazolium compounds (**Be2** and **Be4**) were efficiently internalized by the cells in regular incubation conditions. The emission of **Qe2** was only twice higher than the cell autofluorescence. Both low membrane permeability and low emission quantum yield could have contributed to this effect. The results are in agreement with cytoplasmatic accumulation of quinoliziniums reported earlier in living cells and the colocalization studies further confirm its mitochondria selectivity[22,23]. However, conversely to what was

observed before for fixed cells[45], none of the lipophilic cations appears to efficiently penetrate the nuclear membrane of living cells. Fixation with paraformaldehyde seems to affect the integrity of the nuclear membrane making it more permeable to lipophilic cations. A similar effect was reported for triphenylamines with vinyl branches terminated with pyridinium or benzimidoIium groups[42,50].

Noteworthy, the emission of the red-emitting methoxyphenyl substituted compounds (**Q2** and **B2**) suffers a shift towards shorter wavelengths (blue-shift) upon interaction with the mitochondria (Fig. 6). This effect contrasts to what is typically observed in the popular JC mitochondrial labels where the formation of J-aggregates due to accumulation in the mitochondria leads to a shift of the emission to longer wavelengths (red-shift)[51]. Mitochondria targets based on a rotor mechanism show also red-shifted internal rotation upon interaction with the mitochondria[32].

The observation of this unexpected blue-shift could imply disruption of the conjugation length, but it is not obvious how accumulation at the mitochondrial could lead to such effect. Another alternative is a change in the nature of the emissive state promoted by environment polarity. However, typically, polarity sensitive probes change from a blue emissive locally excited state (LE) to a red-emissive state upon stabilization of the intramolecular charge transfer (CT) in polar environments such as that provided by the mitochondria[27]. Formation of the CT state is accompanied by some degree of structural rearrangement in the so-called twisted CT states (TICT). Indeed, the large Stokes shifts observed in DMSO for all the cationic dyes prepared, and specially for **Q2** and **B2**, suggests that we are in the presence of an emissive CT state. The broad spectra and the lack of vibronic structure supports a significant structural rearrangement upon excitation.

Solvatochromic studies confirmed the existence of two emissive excited states for **Q2** and **B2** (Fig. 6). In low polarity solvents **Q2** and **B2** emit from the LE state at shorter wavelengths and in high polarity solvents they emit from the CT state at longer wavelengths. In some solvents the two states coexist. The highly negative membrane potential of mitochondria is suitable for stabilization of CT states. However, interaction with the mitochondria can also hinder the associated structural change, thus avoiding its formation. The emission spectra collected for the mitochondria-associated dyes do overlap with the spectra of the LE in low polarity solvents (Fig. 6). To further validate this assumption, we prepared a film of **B2** (Fig. 6C) and observed how the emission changed from orange to green as the structural rearrangement was blocked upon drying. This observation confirms that the formation of the CT state is accompanied by a rotamerization process that is hindered in the solid state where only emission from the coplanar LE state is observed. Charge transfer from the methoxy group to the central electron acceptor cation is accompanied by rotation around the C–O bond connecting the methoxy group to the phenyl group that twists the methoxy group out of the molecular plane. This rotational of the methoxy group decouples the non-bonding orbitals of the oxygen atom from the molecular π system, thus blocking the back electron transfer and stabilizing the intramolecular twisted CT state. Interaction with the mitochondria, or alternatively aggregation of the dyes accumulated at the mitochondria, appear to have a similar effect of restriction of the molecular rotation accompanying formation of the CT state whereby only emission of the LE state is observed. In addition, we checked that the spectral profile of emission is not significantly affected by pH nor prolonged UV–Vis irradiation in solution or in cultured cells.

**Two-photon induced mitochondria damage**. Compound **Q2** showed a very efficient light-activated mitochondria damage. This

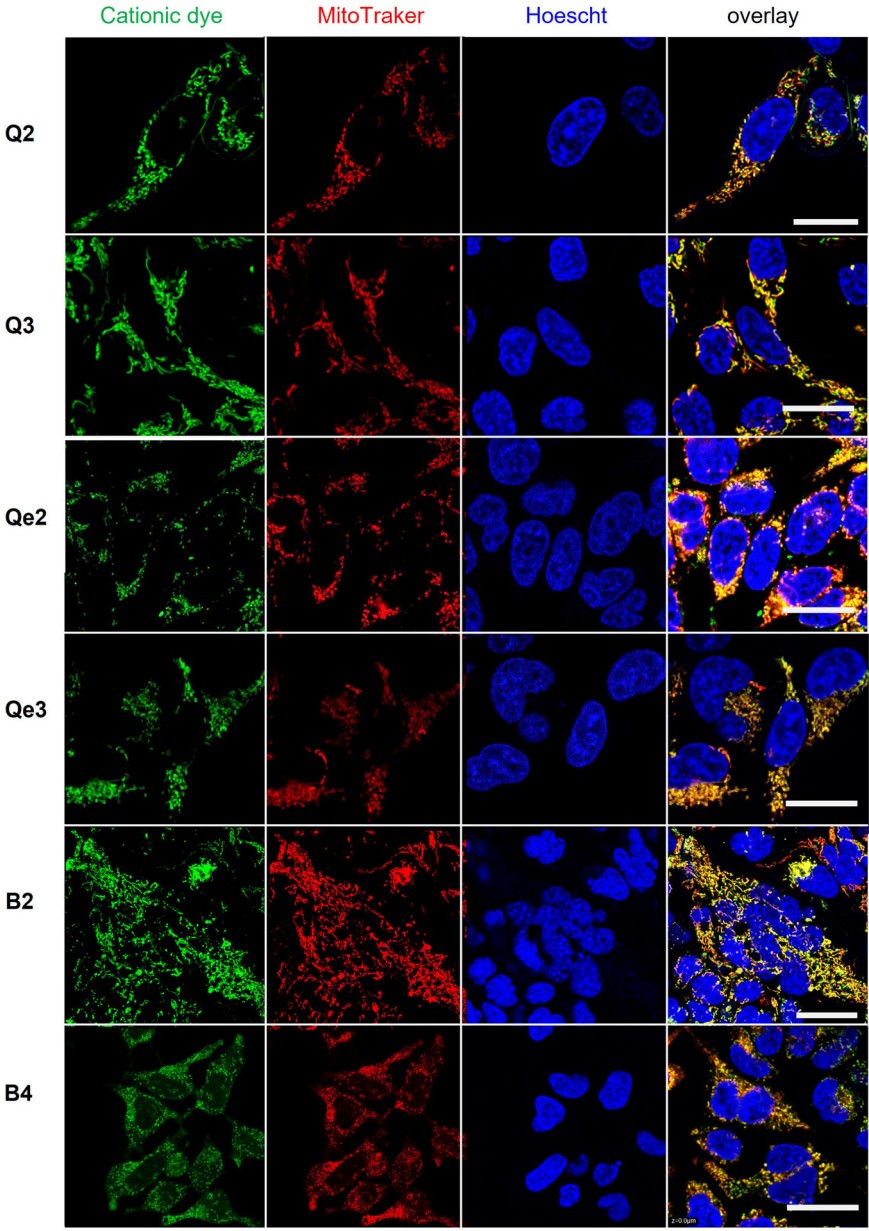

**Fig. 5 Fluorescence microscopy images of HEK 293T cells stained with the quinolizinium and benzimidazolium compounds.** The image shows the compounds **Q2**, **Q3**, **Qe2**, **Qe3**, **B2** and **B4** in the green channel (λexc/em = 458/500-600 nm), the MitoTracker Red in the red channel (λexc/em = 514/600-700 nm) and the Hoechst 33342 in the blue channel (λexc/em = 780/400-500 nm). The overlay of the isolated channels shows the co-localization of the selected compounds with the MitoTracker Red in yellow. Scale bar is 25 μm.

effect is illustrated in Fig. 7 for **Q2** under two-photon excitation at 800 nm with a focused femtosecond laser delivering 8 mW for about 5 min in scanning mode. Irreversible mitochondria swelling and formation of hollow spherical vesicles are already evident shortly after 20 s of irradiation. The overlay of images taken before and after irradiation can further illustrate the effect in Figure S18 of the Photoirradiation effect section of the supplementary information. Similar morphology changes have been recorded upon one-photon irradiation with a CW laser at 458 nm delivering ≈20 μW in a diffraction-limited volume (Fig. S19). Other compounds in this study showed similar effects (Fig. S20) but were not investigated in detail. The mitochondria dynamics under laser irradiation is illustrated in two movies available as supplementary movie 1 and supplementary movie 2 showing that

mitochondria lose their elongated shape upon 5 min of irradiation of **Q2** at 458 and 800 nm, respectively.

Mitochondria with elongated filamentous shape have a high membrane potential while those with round shape and vesicle-like shape are associated with compromised mitochondria of lower membrane potential and eventually cell death[52]. Oxidative stress is one of the factors causing mitochondria permeability transition pore (MPTP) opening that leads to depolarization of mitochondrial membrane potential, mitochondria swelling and rapid impairment of mitochondrial function. To assess the cell death mechanism in the presence of **Q2**, the formation of reactive oxygen species (ROS) upon two-photon and one-photon excitation was further investigated using MitoSoX red, APF and SOSG probes that signal the presence of mitochondrial

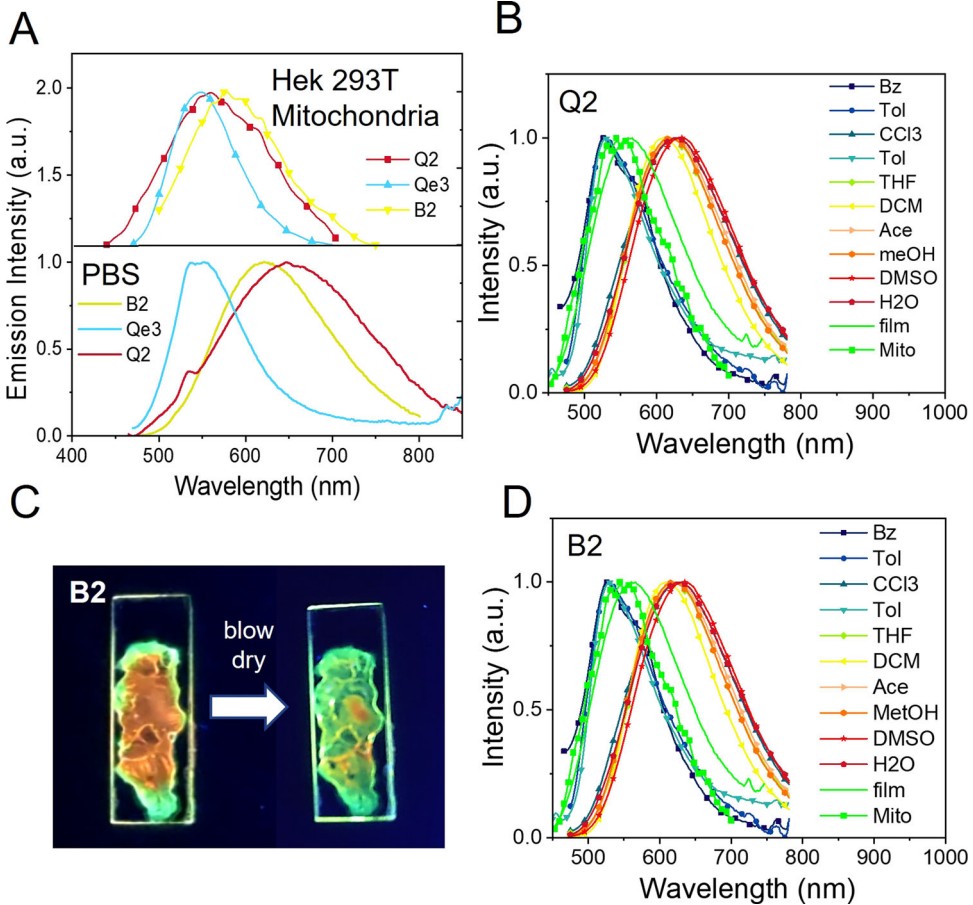

**Fig. 6 Emission spectra of Q2, B2 and Qe3 in cells and in different solvents.** Emission spectra of **Q2**, **B2** and **Qe3** in HEK 293T cells and in PBS at pH7 solution (**A**), solvatochromic studies for the red emitting **Q2** (**B**), photograph taken under UV irradiation showing the emission of a film of **B2** changing from orange to green while it dries (**C**) and solvatochromic studies for the red emitting **B2** (**D**). Solvatochromic studies show the blue-shift in emission in low polarity solvents (Bz, Tol, CHCl₃), in the film and in the mitochondria. In the legend: Bz, benzene; Tol, toluene; CHCl₃, chloroform; DE, diethyl ether; THF, tetrahydrofuran; DCM, dichloromethane; Ace, acetone; MeOH, methanol; DMSO, dimethylsulfoxide; H₂O, water; Mito, mitochondria in HEK293T cells.

superoxide[53], hydroxyl radical[54], and singlet oxygen[55], respectively. Figure 8 shows that all the probes gave a positive identification of the corresponding ROS under irradiation at 800 nm for 20 s.

In Fig. 8, we show the overlay between the green channel of **Q2** and the red channel of the ROS probe. Before irradiation, only the green colour of **Q2** is observed in agreement with the fact that all the probes should be weakly or non-emissive in the absence of ROS. After irradiation there is an evident increase in the intensity of the red channel in the presence of the SOSG and APF and MitoSOx red probes. For MitoSOX red + **Q2**, both dyes are selective to the mitochondria resulting in the appearance of a yellow colour in the overlay due to the overlap between emission in the red (MitoSOX red) and green (**Q2**) channels. The increase in the emission of all the probes studied upon irradiation in the presence of **Q2** can be further confirmed in the intensity profile plots in panels G–I.

The increased intensity of SOSG emission signals the production of singlet oxygen by triplet-triplet energy transfer from the triplet state of **Q2** to molecular oxygen (³O₂) following a Type II photosensitized oxidation reaction. Such mechanism of singlet oxygen production (¹O₂) upon excitation of lipophilic cationic dyes has been documented for acridine orange, methylene blue and toluidine blue that have high quantum yields

of triplet formation, with energies that are high enough to produce singlet oxygen via type II mechanism[56]. Quinolizinium derivatives have been shown to undergo fast S₁ →T₁ intersystem crossing, thus supporting the high yields of singlet oxygen photogeneration in this type of compounds[57]. MitoSOX red can cross the phospholipid bilayer and accumulate in the mitochondrial matrix where it is rapidly and selectively oxidized by mitochondrial superoxide to a highly fluorescent product. Radical ROS, in particular superoxide, can be naturally generated by the mitochondria during oxidative ATP production[58]. The increased emission intensity of MitoSox red upon irradiation in the presence of **Q2** shows that there is an increase in the production of mitochondrial superoxide. It is unlikely for the superoxide to be a product of a type I photosensitized oxidation reaction involving a direct one-electron transfer from **Q2** (an electron-deficient species) to molecular oxygen. Alternatively, the burst of superoxide is likely a consequence of the MPTP opening associated with oxidative stress induced by singlet oxygen photogeneration in the presence of **Q2**. In addition, MPTP opening is known to increase the ROS production by the mitochondria. Thus, there is a virtuous cycle of ROS-induced ROS release through which superoxide, and other radial ROS, can be generated[58]. Additional evidence of such mechanism is provided by the increased emission intensity of the APF probe

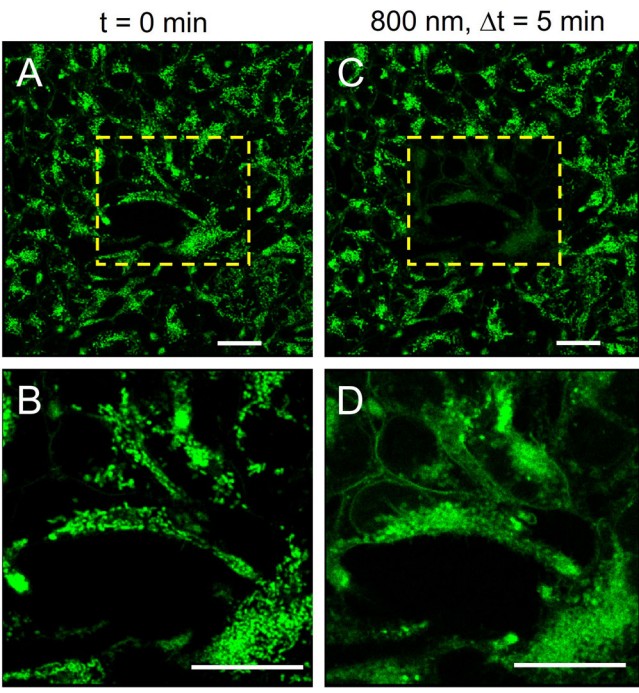

**Fig. 7 Fluorescence microscopy images showing the effect of two-photon excitation (800 nm, 8 mW, 5 min) of HEK293T cells incubated with 1.8 μM of Q2.** The light was focused through a 63 × 1.2 N.A. water immersion objective and the irradiation was done in scanning mode with 400 Hz per line collecting 512 × 512 pixel images. The emission of **Q2** was collected at 500–700 nm. Left panels (**A** and **B**) are images taken before irradiation. The right panels (**C** and **D**) were taken after irradiation. The lower panels are an amplification of the highlighted area in the top panels. Scale bar is 25 μm.

upon irradiation in the presence of **Q2** that signals the production of other radical ROS by the mitochondria, with greater specificity towards hydroxyl radical (OH·).

In the absence of **Q2**, mitochondria integrity was confirmed upon irradiation of HEK293T cells incubated with a control mitochondria stain (Rhodamine 123) using the same excitation conditions for about 3 min (Fig. 8L–M). The overall emission intensity remains unchanged while some intensity redistribution is observed due to mitochondria dynamics (Fig. 8N). We note that for the mitochondria selective probes (MitoSox red and Rhodamine 123), in addition to the effects of irradiation, the random movement of the mitochondria induces a redistribution of peaks and troughs in the intensity profile. To eliminate the effect of mitochondria dynamics and better convey the irradiation effect, Figure S21 in the Supplementary Information shows the distribution of pixel intensity in a wider region of interest (ROI) within the cells for these two probes. Figure S21 confirms that the average pixel intensity in the MitoSox red channel increases up to the saturation level upon irradiation, whereas for the negative control of Rhodamine 123 the distribution of pixel intensity is only slightly shifted towards lower intensities. Most importantly, in the negative control, the mitochondria retain their elongated shape upon irradiation in the absence of **Q2**. Control experiments were also performed in cultured cells incubated with the ROS probes in the absence of **Q2** to ensure that irradiation alone does not induce a change in their emission intensity (Fig. S22). In the absence of irradiation, **Q2** alone in the dark has no effect on the mitochondria shape. The IC$_{50}$ of **Q2** (21 μM) in the dark was estimated to be one order of magnitude larger than the concentration used in the photoirradiation studies.

The production of singlet oxygen ($^1$O$_2$) in the presence of **Q2** was followed also in solution using the same SOSG and APF probes used in the cultured cells (Fig. 8J, K). We note that due to the presence of 1% DMSO in solution coming from the stock solution of **Q2**, the APF probe will only respond to the presence of $^1$O$_2$, as discussed elsewhere[54]. MitoSOX red was not used in solution because it can only be used as a probe for superperoxide in live cells. The singlet oxygen quantum yield of **Q2** was determined to be 0.40 ± 0.05 in PBS at pH7 using Rose bengal (RB) as a reference photosensitizer and SOSG as a probe for $^1$O$_2$. It is noteworthy that the singlet oxygen photogeneration yield inside the cells might be different from that determined in PBS due to the LE nature of the singlet excited state stabilized upon interaction with the mitochondria, as opposed to the CT state stabilized in aqueous solution. Nevertheless, the formation of singlet oxygen upon irradiation of **Q2** has been duly signalled by the increased intensity of SOSG under irradiation in the cell culture. The reported photoinduced mitochondria damage opens the possibility of spatially controlled light actuated mitochondria deactivation using either one-photon excitation in the visible or two-photon excitation in the NIR. Two-photon excitation is particularly interesting because the intrinsic spatial localization of nonlinearly induced processes enables targeting of well-defined subsets of mitochondria while the quadratic dependence of the processes on the excitation power allows for a fine control of the extent of mitochondria impairment.

## Conclusions

In summary, a set of fluorescent mitochondria targets were designed based on dipolar and quadrupolar quinolizinum and benzimidazolium cations. Beyond the use of traditional lipophilic cations such as triphenylphosphonium, pyridinium and indolium, our approach to mitochondria targeting is original as far as molecular design is concern[27]. All the newly reported dipolar compounds were efficiently internalized by the cell and exhibited high mitochondria selectivity. Compound **Q2** shows the highest two-photon brightness with 65 GM (σ$_2$ of 266 GM and ϕ = 0.24). Interaction with the mitochondria leads to an unexpected blue-shift of the emission for the far-red emitting compounds **B2** and **Q2**. Severe mitochondria damage could be observed upon two-photon excitation in the NIR. We envision that these observations could lead to the development of new optical tools to interact with the mitochondria. Considering that targeting of mitochondria is nowadays one of the promising therapeutic routes to eliminate cancer, and organelle-targeted photosensitizers are a new paradigm for efficient photodynamic therapy[29,59–61], the results here presented might also inspire the design of light actuated anti-cancer drugs.

## Methods

**Synthetic procedures**. The procedure to obtain quinolizinium-based dipolar and quadrupolar molecules is described in detail in the Synthesis and Characterization section of the supplementary information and illustrated in Scheme S1. Briefly, although dipolar quinolizinium based chromophores (**Q2** and **Q3**) can be obtained by combining the 2-vinyl quinolizinium hexafluorophosphate with aryl donor units, using the Heck reaction (37-42% yield) following a procedure that was earlier described[62], the corresponding chromophores were obtained in better yields from 2-methylquinolizinium salt as hexafluorophosphate or tetrafluoroborate[63] by simple Knoevenagel condensation (80–82% yield). Quadrupolar molecules, **Qe2** and **Qe3** were obtained from 2,8-dimethylquinolizinium (PF$_6$⁻ or BF$_4$⁻) by a double Knoevenagel condensation on the corresponding aryl aldehyde, in higher yields.

To access D-π-A⁺ and D-π-A⁺-A⁺-π-D structures of the benzimidazolium-based compounds, we first synthetized **B′** and **Be′** precursors. For dipolar derivatives, the synthetic approach starts with readily available 2-methyl benzimidazole, via condensation of orthoesters and 1,2-phenylenediamines under mild reaction conditions in the presence of iodine as an efficient catalyst[64]. For quadrupolars, starting from 2,2′-dimetil-1H, 3′H-5,5′-bibenzimidazol by a ring-forming reaction from 1,1′-bifenil-3,3′,4,4′-tetraamine and condensation with

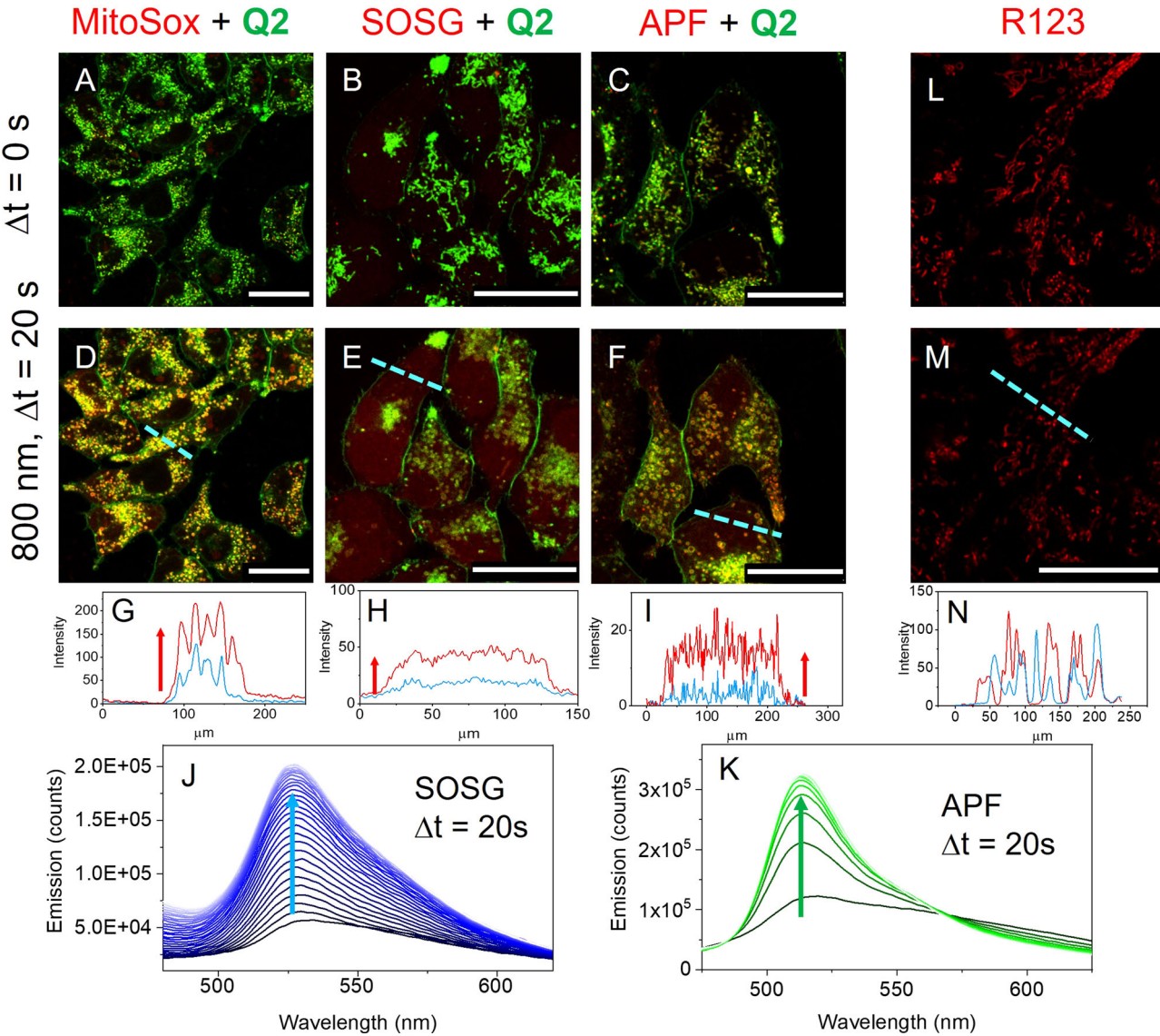

**Fig. 8 Effect of two-photon irradiation (800 nm, 16 mW, 20 s) on HEK293T cells incubated with different ROS probes (MitoSOX red, SOSG and APF) in the presence of Q2 (1.8 μM). A–F** the overlay between the emission of **Q2** (green) and emission of the ROS probes (red) before (**A–C**) and after (**D–F**) irradiation. The irradiation was done in scanning mode with 400 Hz per line at 512 × 512 pixels per scan using a focused beam through a 63 × 1.2 N.A. water immersion objective. The **Q2** channel was recorder under 458 nm excitation by collecting emissions at 485–650 nm. The ROS probe channels were recorded under 514 nm excitation by collecting emission at 570–690 nm (MitoSox red) or 530–650 nm (SOSG and APF). **G–I** are intensity profiles for the ROS probe along the cyan dashed line in (**D–F**). The blue and red profiles correspond to the ROS probe channel before and after irradiation, respectively. The intensity profiles in (**G–I**) show that two-photon excitation increases the intensity of the ROS probe channel. For SOSG and APF the same effect was observed in solution in the presence of **Q2** as illustrated in (**J**) and (**K**). **L–N** are control experiments where the cells were incubated with a mitochondria label (Rhodamine 123) showing that two-photon excitation alone does not produce any change in mitochondria morphology. Scale bar equals to 25 μm.

orthoester, followed by alkylation with methyl iodide in presence of sodium hydride (Scheme S3)[65]. The target compounds **B2** and **B4** were obtained using a classical approach involving a Knoevenagel reaction between **B'** and the corresponding arylaldehydes [4-dimethylamino-benzaldehydes or 4-[(1E)-2-(4-methoxy-phenyl)ethenyl]-benzaldehydes)] in high yields, using pyridine as a base catalyst in methanol at reflux as indicated in Scheme S2a. Analogously, the process was completed via standard Knoevenagel-type condensations of **Be'** with the appropriated benzaldehydes (Scheme S2b). The materials and reagents used are included in Section 1.1 of the supplementary information. A detailed description of the synthetic procedures and the structures of these new chromophores, unambiguously confirmed by their analytical and spectral data are also available in the supplementary information (Section 1.2–1.4 and Figs. S1–S8).

**Optical properties**. The linear absorption spectra were recorded in a JASCO V-540 spectrophotometer. The fluorescence spectra were recorded using a Horiba Jobin Yvon Fluorlog 3-22 Spectrofluorimeter with a xenon lamp of 450 W as excitation source. The fluorescence quantum yields were determined in DMSO using either Fluorescein or Rhodamine 101 as quantum yield standard depending on the overlap of the absorption spectra of the standard and the unknown compound as listed in Table S1 of the Optical properties section in the supplementary information.

The TPA spectra were measured in DMSO by two-photon fluorescence (TPF) using Rhodamine 6G in methanol as standards to account for collection efficiency and pulse characteristics[66]. A modified setup that follows the one described by Xu and Webb was used to estimate the TPA cross-section in the range 710–990 nm region[67]. The two-photon emission (TPE) was measured within a narrow wavelength bandwidth selected by the H20Vis Jobin Yvon monochromator placed at the entrance of a PMC-100-4 photomultiplier tube (Becker and Hickl GmbH). The integrated TPF over the entire emission band was extrapolated using the emission spectra corrected by the detector sensitivity. The excitation source was a Ti:sapphire laser (Tsunami BB, Spectra-Physics, 710–990 nm, 1.7 W, 100 fs, 82 MHz). Solutions in the μM concentration range in DMSO were used. The TPA

cross-section was calculated from the equation:

$$\sigma_2 = \left(\frac{F_2}{\phi C n}\right)_s \left(\frac{\phi C n \sigma_2}{F_2}\right)_{ref} \tag{1}$$

where $F_2$ stands for two-photon induced fluorescence intensity, $\phi$ is the one-photon excited fluorescence quantum yield, $n$ refers to the solution refractive index, $C$ is the concentration and $s$ and ref are relative to the sample and the TPA reference, respectively. The emission intensity dependence of the excitation power was checked. The relative error of the cross-sections values is at most ±20%. To check the overlap between the two-photon and one-photon induced emission, the TPE spectra were also measured in the Leica TCS-SP5 microscope using the same excitation source. In this setup, a short-pass filter with cut-off at 650 nm limits our observation window (see TPA in Figs. S11–S14).

**Singlet oxygen quantum yield**. Reagents and solvents were obtained commercially and used without further purification. Stock solutions of SOSG (Invitrogen™, ThemoFisher) were prepared by dissolving 100 µg in 200 µL of methanol to obtain a stock solution of 825 µM. The solution was stored in the freezer and used within 2 days. Stock solutions of Rose Bengal (RB, Sigma-Aldrich) in PBS (20 µM) and **Q2** in DMSO (1 mM) were kept in the dark and stored in the freezer. The solutions of RB and **Q2** used for singlet oxygen quantum yield determination were freshly prepared in PBS at pH 7.

The quantum yield of photogeneration of $^1O_2$ by **Q2** in PBS was investigated using RB as a reference photosensitizer ($\Phi_\Delta^{RB} = 0.76$ in water) and SOSG as a fluorescent probe for $^1O_2$ probe following the procedure reported by Lin et al. [68] The concentrations of RB and **Q2** varied in the range 0.25–1 µM and 4–15 µM, respectively. The emission of the singlet oxygen probe (SOSG, 6 µM)[68] was measured under excitation at 374 nm in the presence of either **Q2** or RB to determine the rate of $^1O_2$ photogeneration for each concentration. The Xe arc lamp (450 W) of the spectrofluorimeter was used for excitation with the wavelength selected by a monochromator. The excitation slit was fully opened during the measurements. The emission spectra were measured continuously in the 480–620 nm. Each spectrum took 22 s to record. The samples were continuously stirred during measurements and the cuvette was open at the top. Illustrative plots of the increase in SOSG emission under such irradiation conditions are shown in Fig. S23 of the Singlet oxygen quantum yield section of the supplementary information. The $^1O_2$ generation quantum yield of **Q2** was determined using the Eq. (2):

$$\Phi_\Delta^{Q2} = \frac{r_{Q2}/C_{Q2}}{r_{RB}/C_{RB}} \cdot \frac{\varepsilon_{RB}}{\varepsilon_{Q2}} \cdot \Phi_\Delta^{RB} \tag{2}$$

where $C_{Q2}$ and $C_{RB}$ are the concentrations and $\varepsilon_{Q2}$ and $\varepsilon_{RB}$ are the molar absorption coefficients at the excitation wavelength for **Q2** and RB, respectively. $r_{Q2}$ and $r_{RB}$ are the slopes of the plot of the $^1O_2$ generation rate upon photosensitization of **Q2** and RB, respectively, as a function of concentration of the photosensitizer, shown in Figure S23. The good linearity of such plots confirms the viability of the procedure used to estimate the $^1O_2$ photogeneration yield. We also checked that the emission of the APF ROS probe, used as an indicator of the generation of ROS by **Q2** in the cell culture, was increasing upon irradiation of **Q2** in solution. Since **Q2** solution was prepared from a stock solution of DMSO, the APF probes is only signalling the presence of $^1O_2$ giving no additional information about other ROS in solution[54].

**Cell incubation and optical microscopy characterization**. Human embryonic kidney cells 293T (HEK 293T) were cultured in Dulbecco's modified Eagle's medium (DMEM, Catalog number 41966-029) supplemented with 10% fetal bovine serum (Thermo Fisher Scientific, Catalog number 10500-064, heat inactivated), 1% Penicillin-Streptomycin (Thermo Fisher Scientific, catalogue number 15140-122) in a 5% CO$_2$ incubator at 37 ˚C. The cells were grown on ibidi µ-Slide 8 well glass bottom. Prior to the addition of cells, the chambers were coated with poly-L-lysine (Sigma, catalogue number P4707) for at least 30 min, then each well was washed several times with Dulbecco's phosphate-buffered saline (DPBS, Thermo Fisher Scientific, catalogue number 14190-094). The cells were subsequently added to the wells and cultured with DMEM (completed medium for 2 days before each experiment. Solutions of $10^{-3}$ M concentration of the charged compounds in DMSO were used as stock solutions. Several staining protocols were attempted by changing the concentration from 1.75 to 10 µM, generally by the stock solutions in DPBS. Cells were subsequently labelled with the desired dyes and ROS probes (charged compounds, MitoTracker, Hoechst 33342 dye, SOSG, MitoSOX red and APF) and incubated for 30 min at 37 ˚C. The commercial dyes and ROS probes were prepared according to the instructions of the supplier (Thermo Fisher Scientific). After labelling is complete the unbounded dyes were removed by washing the cells with DPBS. The cells were imaged using a laser scanning confocal microscope (Leica TCS-SP5) equipped with a continuous Ar ion laser (Multi-line LASOS® LGK 7872 ML05) and a Ti:sapphire (Spectra-Physics Mai Tai BB, 710–990 nm, 100 fs, 82 MHz). A 63 × 1.2 N.A. water immersion objective was used (HCX PL APO CS 63.0 × 1.20 WATER UV). Typically, 100 × 100 µm images were collected with 1064 × 1064 pixels at a scan rate of 100–400 Hz. Individual image channels were pseudo-coloured with RGB values corresponding to each of the fluorophore emission spectral profiles.

**Cytotoxicity**. Cytotoxicity studies were performed only for the most promising **Q2** and **B2** dyes. The viability assay PrestoBlue was used to measure the cell cytotoxic response after incubation with distinct doses of **Q2** and **B2** dyes. The cells were seeded in 96-well flat-bottomed polystyrene plates with a density of $1 \times 10^4$ HEK 293T cells/well and left to adhere overnight in a CO$_2$ incubator (5%) at 37 °C. After 24 h, cell medium was discarded and replaced with fresh medium containing different concentrations of each compound. HEK 293T cells were then incubated for a period of 24 h at 37 °C in a humidified 5% CO$_2$ incubator. Then, the effect of both compounds in the metabolic activity of HEK 293T cells was determined using a resazurin reduction fluorometric assay. Resazurin, the active compound in PrestoBlue™, PB (Invitrogen, Carlsbad, CA, USA), is a blue dye that can be reduced to a pink fluorescent intermediate, resorufin, by metabolic active cells[69]. Briefly, after 24 h of incubation the cell culture medium was discarded and replaced by fresh DMEM containing 10% (v/v) PB. The resorufin production was monitored by measuring the fluorescence intensity ($\lambda$exc = 530 nm, ($\lambda$em = 590 nm) in a microplate reader (BMG Labtech, Polar Star Optima) at 37 ˚C. Gain adjustment of the equipment was performed using a positive control, i.e., a sample containing the fully reduced resazurin form obtained after autoclaving for 15 min cell culture medium with 10% (v/v) PrestoBlue, according to with manufacture instructions. The negative control of the assay contained 10% (v/v) resazurin in cell-free medium. The percentage of metabolically active cells/viability was expressed as the percentage of resazurin reduction relative to the reduction measured for the untreated sample after correcting the data with the negative control. The experiments were performed independently in duplicates. A monoexponential decay was fitted to the relative changes detected in the % of cell viability by non-linear regression. The respective dye IC$_{50}$ was determined as IC$_{50}$ = (ln 2)/$k$, where $k$ is the decay constant of the % of cell viability. Data analysis was carried out using Graphpad software. The IC50 in the dark was estimated to be 7 µM for **B2** and 21 µM for **Q2**. For **Q2**, this concentration is one order of magnitude higher than the 2 µM used in the mitochondria targeting and two-photon induced mitochondria damage experiments in live animal cells.

**Reporting summary**. Further information on research design is available in the Nature Research Reporting Summary linked to this article.

## Data availability
All data needed to evaluate the conclusions of this study are presented in this article, the Supplementary Information and/or the Supplementary Movie 1 and Supplementary Movie 2. Additional data related to this study are available from the corresponding authors on reasonable request.

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

## Acknowledgements

Authors gratefully acknowledge the financial support from Fundação para a Ciência e a Tecnologia (FCT), European Union, QREN, FEDER and COMPETE for funding (PTDC/NAN-MAT/29317/2017, PTDC/QUI-QFI/29319/2017 or LISBOA-01-0145-FEDER-029319, LA/P/0056/2020 and UIDB/00100/2020) and from the Spanish Ministerio de Ciencia y Competitividad (MINECO/ CTQ2017-85263-R) and Instituto de Salud Carlos III (ISCIII RETICREDINREN RD16/0009/0015).

## Author contributions

A.C. and E.M. are responsible for the conceptual development of the work. J.R., J.V., and A.C. performed the synthesis and corresponding data analysis; I.M. and A.S. made the optical characterization and related data analysis together with E.M. and J.M.; S.P. and E.M. performed the in vitro experiments and analyzed the microscopy data. All the authors participated in discussion of the data and the writing of the paper.

## Competing interests

The authors declare no competing interests.
