## [Peer Review File · Communications Chemistry]

Reviewers' comments:

Reviewer #1 (Remarks to the Author):

In this paper, the authors reported the synthesis of two-photon activated and mitochondria targeted dyes based on dipolar and quadrupolar quinolizinium and benzimidazolium cations. Some of the compounds show a high affinity towards mitochondria and an efficient light-induced mitochondria damage upon two-photon excitation in the NIR range. The interaction of dyes with DNA increasing their quantum yields in the nucleus, and aggregation or hindered rotation increasing the quantum yield at the mitochondria. The reported photoinduced mitochondria damage opens the possibility of spatially controlled light actuated mitochondria deactivation using either one-photon excitation in the visible or two-photon excitation in the NIR, which is very interesting. In general, the manuscript reports some interesting new results and can be recommended to be published after clarifying the following issues :

1. "To further validate this assumption, we prepared a film of B2 (Figure 6C) and observed how the emission changed from orange to green as the structural rearrangement was blocked upon drying." (14 page, 271 line). The author attributed the color change to the structural rearrangement. We recommend further discussion for the structural rearrangement and molecular aggregation.
2. There are two types of anion (PF6⁻ or I⁻) in these two-photon absorption dyes. Do the anions affect the two-photon absorption cross-section of the cation parts? Please clarify.
3. More discussion regarding the structure-property relationships is needed for the benefits of readers. Besides, theoretical calculations are recommended to gain insights to the structure-property relationships.

Reviewer #2 (Remarks to the Author):

In this manuscript, the authors report a molecular system based on quinolizinium and benzimidazolium that utilizes their highly efficient two-photon absorption properties and singlet oxygen production performance. It is noteworthy that the molecules have mitochondrial targeting properties and that some of their optical properties change upon interaction within the mitochondria. As described by the authors, the development of molecules with intracellular organelle-targeting properties is an area of research that has received particular attention in recent years.

Since the molecular performance (especially Q2) and cellular responses are presented with appropriate data, I may recommend the publication of the article in this journal. However, I feel that the following points are unclear or questionable, and the authors should account them before publication.

- (1) It is not clear from this manuscript why quinolizinium and benzimidazolium show relatively high singlet oxygen production performance. The reason should be explained with appropriate citation(s) or data during the introduction or discussion.
- (2) Regarding the discussion of Figures 6B and 6D from line 271, the authors should clearly state what we can understand from the film sample data and can conclude. (It should not be left to the reader to interpret.)
- (3) The authors sometimes say "all the compounds" but do not provide data for all the compounds

sometimes. For example, in line 277, "some degree of light activated mitochondria damage.", no data on damage is given for all the compounds. Such inaccurate statements should be corrected overall.

(4) In Supporting Information, errors are sometimes found. These are not inherently problematic from a scientific perspective but should be checked and corrected again before publication. For example, there is a mismatch in the number of significant digits, and in Section 1.3.2 Synthesis of dipolar D- π -A⁺ benzimidazoliums, "mixture reaction" should be "reaction mixture".

(5) For example, in Figure 8, the cyan line data is picked up, but as the authors show in the video, the cells (mitochondria) move a lot during 20 seconds, so such partial data extraction is inappropriate. It would be more appropriate to show the fluorescence changes in a wider area, or even in an entire cell.

(6) The data of MitoSOX and APF are shown to elucidate the mechanism of cell death, but what can be concluded from these data is not clearly stated (though it is implied). It should be also explained why the use of MitoSOX is effective. (I can understand it, but it is difficult for the general reader.)

(7) Although the efficiency of singlet oxygen generation was verified in PBS solution, the effect of the optical change on the efficiency of singlet oxygen generation in mitochondria, where the optical properties of molecules are changing, should be described. Inference may be acceptable.

(8) The description in the caption of the video is unclear. "not shown" should be added; "No damage is observed when irradiation is done in the absence of the photosensitizer quinolinium (not shown)".

Once the above points have been corrected as much as possible, this paper is ready for publication.

Point-by -point reply to the reviewers

We thank the reviewers for their insightful comments. With their help, we believe that the revised version has been considerably improved with a more in-depth discussion of the data. In the following there is a point-by-point reply (in blue) to the Reviewers' comments (in black).

Reviewer #1 (Remarks to the Author):

In this paper, the authors reported the synthesis of two-photon activated and mitochondria targeted dyes based on dipolar and quadrupolar quinolizinium and benzimidazolium cations. Some of the compounds show a high affinity towards mitochondria and an efficient light-induced mitochondria damage upon two-photon excitation in the NIR range. The interaction of dyes with DNA increasing their quantum yields in the nucleus, and aggregation or hindered rotation increasing the quantum yield at the mitochondria. The reported photoinduced mitochondria damage opens the possibility of spatially controlled light actuated mitochondria deactivation using either one-photon excitation in the visible or two-photon excitation in the NIR, which is very interesting. In general, the manuscript reports some interesting new results and can be recommended to be published after clarifying the following issues :

1. "To further validate this assumption, we prepared a film of B2 (Figure 6C) and observed how the emission changed from orange to green as the structural rearrangement was blocked upon drying."(14 page, 271 line). The author attributed the color change to the structural rearrangement. We recommend further discussion for the structural rearrangement and molecular aggregation.

REPLY: In the revised manuscript we have clarified this point by adding the following discussion to page 16: "This observation confirms that formation of the CT state is accompanied by a rotamerization process that is hindered in the solid state where only emission from the coplanar LE state is possible. Charge transfer from the methoxy group to the central electron acceptor cation is accompanied by rotation around the C-O bond connecting the methoxy group to the phenyl group that twists the methoxy group out of the molecular plane. This rotation of the methoxy group decouples the non-bonding orbitals of the oxygen atom from the molecular π system, thus blocking the back electron transfer and stabilizing the intramolecular CT state. Interaction with the mitochondria, or alternatively aggregation of the dyes accumulated at the mitochondria, appear to have a similar effect of restriction of the molecular rotation as observed in the solid-state film whereby only emission of the LE state is observed."

2. There are two types of anion (PF₆⁻ or I⁻) in these two-photon absorption dyes. Do the anions affect the two-photon absorption cross-section of the cation parts? Please clarify.

REPLY: Indeed, we have observed that the change in the counterion from PF₆⁻ to BF₄⁻ in Qe3 has a significant effect on the linear and non-linear absorption cross-section, both decreasing by more than one order of magnitude, as documented in Table S2. The shape of the absorption and emission spectra was not affected by the anion exchange, nor was the emission quantum yield. Even though we have not performed a very systematic study, the anion exchange effect appeared to be stronger in compounds with a weaker push-pull effect. For the quinolizinium

derivatives with a strong dimethylamine electron donor (VDMA in ref 4) no anion exchange effect was observed, whereas in the equivalent compound with a weaker methoxy electron donor (VMOP in ref 4) the exchange of PF_6^- by BF_4^- resulted in a decrease of the cross-section by a factor of 2, and in the more extended Qe3 structure the drop was larger than one order of magnitude.

REPLY: Thus, to avoid effects related with the nature of the counterion we have kept the counterion constant within each series (quinolizinium with PF_6^- and benzimidazolium with I^-). Due to the strong electron donor nature of the dimethylamine donor in the benzimidazolium series no strong counterion effect is expected.

Section 2. Optical Properties of the ESI was revised to include a more extended note about the anion effect along the lines presented above.

3. More discussion regarding the structure-property relationships is needed for the benefits of readers. Besides, theoretical calculations are recommended to gain insights to the structure-property relationships.

REPLY: Geometry optimization and vertical transition energies have been calculated using CAM-B3LYP/6-311+G(d,p) level of approximation. The wavelengths for the lowest energy transitions, which are also the strongest transitions in the visible, have been included in Table S2. The simulated spectra and frontier molecular orbitals involved in the lowest energy transitions are shown in Figs S9 and S10. The discussion on structure property relationships has been improved in section "Linear optical properties" with the help of computational calculations.

Reviewer #2 (Remarks to the Author):

In this manuscript, the authors report a molecular system based on quinolizinium and benzimidazolium that utilizes their highly efficient two-photon absorption properties and singlet oxygen production performance. It is noteworthy that the molecules have mitochondrial targeting properties and that some of their optical properties change upon interaction within the mitochondria. As described by the authors, the development of molecules with intracellular organelle-targeting properties is an area of research that has received particular attention in recent years.

Since the molecular performance (especially Q2) and cellular responses are presented with appropriate data, I may recommend the publication of the article in this journal. However, I feel that the following points are unclear or questionable, and the authors should account them before publication.

(1) It is not clear from this manuscript why quinolizinium and benzimidazolium show relatively high singlet oxygen production performance. The reason should be explained with appropriate citation(s) or data during the introduction or discussion.

REPLY: The mechanism whereby quinolizinium and benzimidazolium cations can operate as PS has been elucidated in page 20.

"The increased intensity of SOSG emission signals the production of singlet oxygen by triplet-triplet energy transfer from the triplet state of Q2 to molecular oxygen ($^3\text{O}_2$) following a Type II

photosensitized oxidation reaction. Such mechanism of singlet oxygen production ($^1\text{O}_2$) upon excitation of lipophilic cationic dyes has been documented for acridine orange, methylene blue and toluidine blue that have high quantum yields of triplet formation, with energies that are high enough to produce singlet oxygen via type II mechanism.⁵⁶ Quinolizinium derivatives have been shown to undergo fast $S_1 \rightarrow T_1$ intersystem crossing, thus supporting the high yields of singlet oxygen photogeneration in this type of compounds.⁵⁷

(2) Regarding the discussion of Figures 6B and 6D from line 271, the authors should clearly state what we can understand from the film sample data and can conclude. (It should not be left to the reader to interpret.)

REPLY: We have completed the discussion about the film sample by including the following sentence on page 14: "This observation confirms that formation of the CT state is accompanied by a rotamerization process that is hindered in the solid state where only emission from the coplanar LE state is possible. Charge transfer from the methoxy group to the central electron acceptor cation is accompanied by rotation around the C-O bond connecting the methoxy group to the phenyl group that twists the methoxy group out of the molecular plane. This rotation of the methoxy group decouples the non-bonding orbitals of the oxygen atom from the molecular π system, thus blocking the back electron transfer and stabilizing the intramolecular CT state. Interaction with the mitochondria, or alternatively aggregation of the dyes accumulated at the mitochondria, appear to have a similar effect of restriction of the molecular rotation as observed in the solid-state film whereby only emission of the LE state is observed."

(3) The authors sometimes say "all the compounds" but do not provide data for all the compounds sometimes. For example, in line 277, "some degree of light activated mitochondria damage.", no data on damage is given for all the compounds. Such inaccurate statements should be corrected overall.

REPLY: Light activated mitochondria damage was observed for all the compounds with an efficient cellular uptake, but we did not document the effect in a systematic form for all the compounds. Instead, we opted to focus on the effect in Q2 where it appeared to be stronger. In the revised manuscript we have made a clearer statement and give more example of light activated mitochondria damage besides Q2 (B2 e Qe3). We feel that it would also not be entirely correct to omit this observation even though it was not systematically investigated.

The sentence in the original line 277 was corrected: "Compound **Q2** showed a very efficient light activated mitochondria damage."

Figure S20 was introduced in the ESI showing the effect of irradiation in the visible with the CW lamp in the presence of B2 and Qe3, in the mitochondria morphology.

In page 17, line 321 we included "Other compounds in this study showed similar effects (Figure S20), but were not investigated in detail."

(4) In Supporting Information, errors are sometimes found. These are not inherently problematic from a scientific perspective but should be checked and corrected again before publication. For example, there is a mismatch in the number of significant digits, and in Section 1.3.2 Synthesis of dipolar D- π -A+ benzimidazoliums, "mixture reaction" should be "reaction mixture".

REPLY: In the revised version of ESI, the grammar and spelling were checked, and the number of significant digits verified.

(5) For example, in Figure 8, the cyan line data is picked up, but as the authors show in the video, the cells (mitochondria) move a lot during 20 seconds, so such partial data extraction is inappropriate. It would be more appropriate to show the fluorescence changes in a wider area, or even in an entire cell.

REPLY: We note that the random movement of the mitochondria affects only the intensity profiles of the dyes that accumulate at the mitochondria, which is the case of MitoSox red (panel G) and the negative control of R123 (panel N). In these two panels, the peaks and troughs in the intensity profiles before and after irradiation do not coincide due to the random movement of the mitochondria. Nevertheless, it is clear that the average intensity in panel G increases upon irradiation while that in panel N remains roughly the same.

To eliminate the effect of random movement of the mitochondria and convey the irradiation effect in a more appropriate form, for these two probes (MitoSox red and R123), we included Figure S21 in the ESI that shows the distribution of pixel intensity in a wider region of interest (ROI) within the cells. The figure shows that the average pixel intensity in the MitoSox channel is around 41, on a scale of 1-255, before irradiation whereas after irradiation most of the pixels are saturated. For R123, Figure S21 shows that the distribution of pixel intensity is only slightly shifted towards lower intensities after irradiation.

A comment was introduced in the discussion of Fig 8 starting on page 22, Line 411.

(6) The data of MitoSOX and APF are shown to elucidate the mechanism of cell death, but what can be concluded from these data is not clearly stated (though it is implied). It should be also explained why the use of MitoSOX is effective. (I can understand it, but it is difficult for the general reader.)

REPLY: The evidence provided by MitoSOX and APF about the cell death mechanism has been discussed in more detail in page 21:

MitoSOX can cross the phospholipid bilayer and accumulate in the mitochondrial matrix where it is rapidly and selectively oxidized by mitochondrial superoxide to a highly fluorescent product. Radical ROS, in particular superoxide, can be naturally generated by the mitochondria during oxidative ATP production. The increased emission intensity of MitoSox upon irradiation in the presence of Q2 shows that there is an increase in the production of mitochondrial superoxide. It is unlikely for the superoxide to be a product of a type I photosensitized oxidation reaction involving a direct one-electron transfer from Q2 (an electron deficient species) to molecular oxygen. Alternatively, the burst of superoxide is likely a consequence of the MPTP opening associated with oxidative stress induced by singlet oxygen photogeneration in the presence of Q2. In addition, MPTP opening is known to increase the ROS production by the mitochondria. Thus, there is a virtuous cycle of ROS-induced ROS release through which superoxide, and other radical ROS, can be generated.⁵⁸ Additional evidence of such mechanism is provided by the increased emission intensity of the APF probe upon irradiation in the presence of Q2 that signals the production of other radical ROS by the mitochondria, with greater specificity towards hydroxyl radical (OH·).

(7) Although the efficiency of singlet oxygen generation was verified in PBS solution, the effect of the optical change on the efficiency of singlet oxygen generation in mitochondria, where the optical properties of molecules are changing, should be described. Inference may be acceptable.

REPLY: Indeed, triplet photogeneration yield might be quite different inside the cell as compared to the PBS solution. Due to the complexity of the molecular environment, it is also quite difficult to predict how the change in the nature of the singlet excited state can affect the triplet generation yield inside the cells. Nevertheless, we did verify experimentally using the SOSG probe that indeed Q2 can efficiently generate singlet oxygen. In the revised manuscript we have included a comment regarding this issue on page 23:

“It is noteworthy that the singlet oxygen photogeneration yield inside the cells might be different from that determined in PBS due to the LE nature of the singlet excited state stabilized upon interaction with the mitochondria, as opposed to the CT state stabilized in aqueous solution. Nevertheless, formation of singlet oxygen upon irradiation of Q2 has been duly signaled by the increased intensity of SOSG under irradiation in the cell culture. “

(8) The description in the caption of the video is unclear. "not shown " should be added; “No damage is observed when irradiation is done in the absence of the photosensitizer quinolizinium (not shown)".

REPLY: Corrected as suggested.

REVIEWERS' COMMENTS:

Reviewer #1 (Remarks to the Author):

The authors have revised their manuscript carefully and addressed the issues from the referees. I have no further question. The manuscript can be accepted for publication.

Reviewer #2 (Remarks to the Author):

The authors have responded well to the reviewers' questions, and the contents are appropriate. The manuscript can be accepted.